# Buccal Bone Thickness in Anterior and Posterior Teeth—A Systematic Review

**DOI:** 10.3390/healthcare9121663

**Published:** 2021-11-30

**Authors:** Diana Heimes, Eik Schiegnitz, Robert Kuchen, Peer W. Kämmerer, Bilal Al-Nawas

**Affiliations:** 1Department of Oral and Maxillofacial Surgery, University Medical Center Mainz, Augustusplatz 2, 55131 Mainz, Germany; Eik.Schiegnitz@unimedizin-mainz.de (E.S.); Peer.kaemmerer@unimedizin-mainz.de (P.W.K.); Al-Nawas@uni-mainz.de (B.A.-N.); 2Institute for Medical Statistics, Epidemiology and Informatics, University Medical Center of the Johannes-Gutenberg-University Mainz, 55131 Mainz, Germany; Rokuchen@uni-mainz.de

**Keywords:** dental implant, tomography, dental implant loading, immediate, alveolar bone, buccal bone thickness

## Abstract

(1) Background: Immediate dental implant placement has been a subject of great interest over the last decade. Here, information regarding the anatomy and bone thickness of the jaw prior to dental implant placement is crucial to increase the surgery’s success and the patient’s safety. The clinical premises for this approach have been controversially discussed. One of those heavily discussed premises is a buccal bone thickness of at least 1 mm thickness. This meta-analysis aims to systematically review buccal bone thickness (BBT) in healthy patients. Thus, the feasibility of immediate dental implant placement in daily practice can be assessed. (2) Methods: A search in the electronic databases was performed to identify articles reporting on BBT that was measured by computed tomography in adults. (3) Results: We were able to find 45 studies, including 4324 patients with 25,452 analyzed teeth. The analysis showed a BBT at the alveolar crest of 0.76 ± 0.49 mm in the maxillary frontal and of 1.42 ± 0.74 mm in the maxillary posterior region. In the mandible, the average measured values were similar to those in the maxilla (front: 0.95 ± 0.58 mm; posterior: 1.20 ± 0.96 mm). In the maxillary frontal region 74.4% and in the mandibular frontal region 61.2% of the crestal buccal bones showed widths <1 mm. (4) Conclusions: In more than 60% of the cases, the BBT at the alveolar crest is <1 mm in maxillary and mandibular frontal regions. This anatomic data supports careful pre-surgical assessment, planning of a buccal graft, and critical selection of indication for immediate implant placement, especially in the maxillary and mandibular frontal and premolar region.

## 1. Introduction

Intraosseous dental implants are considered a reliable method for replacing missing teeth and restoring a patient’s masticatory function. With different protocols for a variety of indications discussed in the literature, information regarding the anatomy and bone thickness of the jaw prior to dental implant placement is crucial to increase a surgery’s success and a patient’s safety. For each protocol, data concerning survival time, success rates, and peri-implant bone loss are required to allow for a comparison between different options and to be able to assess the success to be expected in each patient.

Especially when immediate implant placement is required, a detailed analysis of the present clinical conditions is needed. Different types of implant placement protocols are defined: Type 1 includes immediate implant placement (with immediate restoration, early loading and conventional loading), Type 2 includes early placement with soft tissue healing (4–8 weeks), Type 3 includes early placement with partial bone healing (12–16 weeks), and Type 4 includes late placement (>6 months) [1,2,3,4]. Immediate implant placement is defined as implant insertion into the socket on the same day as tooth extraction and should be considered in the presence of patient-centered advantages, such as an aesthetic outcome or a reduced morbidity. Survival rates for immediate placement range from 87 to 100% depending on the type of loading protocol [4,5,6,7]. As immediate dental implant placement is a complex surgical procedure, it is recommended to only be performed by clinicians with a high level of experience and in the presence of the following clinical conditions: Intact socket walls, thick soft tissue, no acute infection at the site, bone apical and lingual to the socket, insertion torques of 25 to 40 Ncm or implant stability quotient (ISQ) value >70, patient compliance, and a facial bone wall that is at least 1 mm thick [4]. Immediate implant placement is an attractive technique, since it enables the immediate restoration of aesthetics and at the same time reduces the number of surgical steps, significantly reducing the time needed for dental restauration. However, it is still premature to consider this procedure as a general standard in implantology [8]. Several animal and human studies [9,10,11] have shown that the alveolar ridge undergoes an unavoidable remodeling process after tooth extraction that leads to a reduction of the bone dimension. The resorption probably results from the interruption of the blood supply together with a tendency to a higher osteoclastic activity [12]. In particular, the anterior maxilla consists of a very thin buccal bone that is reported to be made up of bundle bone [13] and considered to be part of the periodontium, which is why it is reabsorbed after tooth extraction [9]. Furthermore, bone resorption is reported to be much greater at the buccal aspect of the alveolar bone [9]. This renders the need for predicting the degree of future bone loss after tooth extraction [14]. Considering that immediate implant placement gained popularity within the last few years, a critical analysis is needed to evaluate the number of cases fulfilling the clinical conditions–especially in consideration of a reported mean bone loss of 7.5 mm in height when the mean buccal bone thickness is <1 mm compared to a loss in height of just 1.1 mm when buccal bone thickness is ≥1 mm [11].

This study aims to assess the buccal bone thickness of healthy people using original articles on this topic, which contain data of buccal bone thickness measured by (cone beam) computed tomography in people of different ethnicity, gender, and age. Firstly, the mean and standard deviation of the buccal bone thickness (BBT) at different regions in the jaw will be calculated. Secondly, the approximate proportions of BBTs that are below the crucial threshold of 1 mm will be determined. The data obtained will thus be used to evaluate the number of cases fulfilling the <1 mm conditions and identify sites in which cautious consideration of feasibility is mandatory.

## 2. Materials and Methods

### 2.1. Search Strategy

An electronic search was conducted in PubMed (by National Center for Biotechnology Information, U.S. National Library of medicine) and Medline from the earliest records up to December 2020. A search command consisting of the following terms was used: alveolar/buccal/facial/bundle ± bone/plate/shelf/crest/ridge ± thickness/width AND tooth, teeth, maxilla*, mandib*, incisor, canine, premolar, molar AND cone beam computed tomography, computed tomography (CT), cone-beam computed tomography (CBCT), tomography, computed tomography, CT imaging, CBCT imaging (search commands are displayed in Appendix A). The search was limited to human studies; accordingly, the filter “humans” in the “species” section in PubMed was used as a filter to further narrow the search. The full search strategies for all databases, including any filters and limits used are presented in Appendix A.

### 2.2. Selection Process

All articles were screened by title and abstract. Potential articles were examined by full text and checked for eligibility.

### 2.3. Eligibility Criteria

The inclusion criteria were:Human randomized clinical trials, nonrandomized clinical trials, cohort studies, case-control studies, and case series (prospective or retrospective);Studies reporting BBT in healthy, adult patients as dimension between tooth and outer border of the buccal bone;Studies measuring bone thickness by (cone beam) computed tomography.

The exclusion criteria were:Case reports, systematic and narrative reviews, animal studies, human cadaver studies, editorial, and doctoral theses;Patients with dental implants;Patients with periodontitis, cleft palate, osteoporosis, dysgnathia, skeletal malocclusion, post-augmentation, post-extraction, and impacted teeth;Languages other than English, German, French, and Chinese.

### 2.4. Data Items

The following pieces of information were extracted from the included articles: Author, title, year, number of patients included within the study (separated in men and women), geographic location, number and type of teeth analyzed within the study, measurement method, site of measurement, and BBT at different sites (Table 1).

### 2.5. Risk of Bias Assessment

In order to assess the quality of the articles analyzed within this review, a method similar to the one described by Vignoletti et al. [15] was chosen. The studies were inspected regarding the following criteria: Randomization, blinding of the examiner, definition of inclusion and exclusion criteria, adequate number of patients/teeth analyzed, conduction of conflict of interest, and funding. The studies were then categorized into groups of high, moderate, and low risk of bias. If one of the criteria was not met, a moderate risk of bias was assumed. In the case of two or more unmet criteria, the risk of a bias was considered.

### 2.6. The Detailed Approach

As can be observed in Table 1, the 45 reviewed papers originally contained information on 96 different measurement points, which differ with regard to their position in the jaw (maxilla/mandibular), to their tooth group (incisor/canine/premolar/molar) and to their distance to the alveolar crest. While some measurement points were only provided data by a single paper, others were included in up to 19 studies. Overall, this results in a total of i=1,…, 380 distinct measurement-point samples.

We decided to summarize the information on these 96 measurement points regarding two different dimensions. One dimension relates to the distance from the alveolar crest. One can distinguish among measurements taken between the apex and the surface of the buccal bone (apical), at 4 to 9 mm (medial) and at 1 to 3 mm (crestal) apically to the alveolar crest. On the other hand, measurements were categorized with respect to the tooth group into the following six classes: Maxillary front teeth (central incisor to canine), maxillary premolars, maxillary molars, mandibular front teeth (central incisor to canine), mandibular premolars, and mandibular molars. Combining these two dimensions results in a total of 18 aggregated regions.

Based on this categorization the two goals of this meta-analysis are:
1.To provide the mean ${\bar{x}}_{r}$ and the standard deviation $s_{r}$ that are obtained by pooling all measurement points pertaining to the same aggregated region$\;r\in \left\{1,\ldots ,18\right\}$.2.To approximate the 18 distributions, from which the respective BBTs are assumed to have been generated, and thereby estimate the proportions $\pi_{1}^{<1},\ldots ,\;\pi_{18}^{<1}\;$of BBTs that are smaller than 1 mm.

The second task is complicated by the fact that most papers only cited the mean and standard deviations of their measured values. Therefore, this goal can only be achieved by making distributional assumptions for the BBTs and by simulating values from those assumed distributions.

### 2.7. Statistics

Goal (1) can be achieved by using basic statistics. The pooled mean ${\bar{x}}_{r}$ of one aggregated region r simply corresponds to the weighted average of all measurement-point samples i assigned to that region:(1)$${\bar{x}}_{r}=\frac{\sum_{i=1}^{380}I_{i}^{r}n_{i}{\bar{x}}_{i}}{\sum_{i=1}^{380}I_{i}^{r}n_{i}}$$

$\mathrm{Here},\;n_{i}$ denotes the number of patients who were included in the study that measurement-point sample i was extracted from. $I_{i}^{r}$ is an indicator function that equals 1, if sample i belongs to aggregated region r, and 0 otherwise. The standard deviation of all values assigned to aggregated region r, on the other hand, can be obtained by [16]:(2)$$s_{r}={\left(\frac{\sum_{i=1}^{380}I_{i}^{r}n_{i}\left(s_{i}^{2}-{\bar{x}}_{i}^{2}\right)}{\sum_{i=1}^{380}I_{i}^{r}n_{i}}-{\bar{x}}_{r}^{2}\right)}^{\frac{1}{2}}$$

Objective (2), on the other hand, cannot be attained using basic statistical methods, but a more intricate simulation study has to be applied. Based on sample sizes, the means, and the standard deviations of the BBTs that were given in the 45 included papers, almost 10 million BBT-values were sampled from patient-specific multivariate gamma distributions. Based on these sampled values, the value ranges of the BBTs at the different aggregated regions can be approximated. $\pi_{1}^{<1},\ldots ,\;\pi_{18}^{<1}\;$can then be simply determined for each aggregated region. The detailed mathematical approach is elucidated in Appendix B.

### 2.8. Research Reporting Guidelines

The authors are stating compliance with the PRISMA guidelines 2020 for systematic reviews.

## 3. Results

### 3.1. Selection and Data Collection Process

Two independent reviewers (conventional double-screening) identified a total of 1679 (after duplicated removed) records through database searching (P.W.K., D.H.). The search yielded 353 articles that were screened for title and abstract. The reviewers excluded 245 articles from further processing. A total of 108 full-text articles were further checked for eligibility. Fifty-seven articles were excluded after a full-text review and three articles during data extraction [17,18,19] due to a lack of data within the article and no answer on data request. A total of 45 articles were included in the review for qualitative synthesis (Figure 1 and Table 1). If articles met the inclusion criteria, they were included in the quantitative and qualitative data analysis. All selection steps were performed by two persons independently of each other. Discrepancies were resolved by discussion. In the case of discrepant judgements, a third author (E.S.) was involved.

### 3.2. Study Selection

A total of 4324 patients with 25,452 teeth were analyzed in the global studies. Included were studies published between 2005 and 2020. Values were measured by CBCT or CT scan. Measurement points were all teeth between central incisors and second molars within the upper and lower jaw at different heights.

Some articles were excluded after full-text review for the following reasons: Lack of data referring to BBT [65,66,67,68], measurement not by (CB)CT scan [69] patients matching the exclusion criteria [19,70,71,72,73,74,75,76,77,78,79,80,81,82,83,84], patients with implants [85], patients without teeth [86], data not displayed as mean and SD [56], site or kind of measurement not adequate [19,62,64,77,87,88,89,90,91,92,93,94,95,96,97,98,99,100,101,102,103,104,105,106,107,108,109,110,111,112,113,114], study design not clear [65,115,116], and lack of data and no response on data retrieval [117].

### 3.3. Risk of Bias in Studies

Five studies reported on blinding. All articles answered an appropriate and clearly focused question. Only 6 studies did not exactly define the inclusion criteria, whereas exclusion criteria were stated in 38/45 studies. Conflicts of interest were named in 33/45 studies and the source of funding was given in 20/45. A total of 33 studies claimed no conflict of interest. The data assessment showed one study with a low risk, 10 with a moderate risk, and 34 studies with a high risk of bias (see Appendix A).

### 3.4. Buccal Bone Thickness

First, using (1) and (2) ${\bar{x}}_{r}$ and $s_{r},\;$the means and the standard deviations of the BBTs of the 18 aggregated regions were calculated (Table 2). In the maxillary frontal region, the crestal BBT was 0.76 ± 0.49 mm, while the BBT apical to the alveolar crest and at the apex of the radix was 0.84 ± 0.56 mm and 1.46 ± 0.98 mm, respectively. More posteriorly located maxillary teeth showed higher values. The BBT of maxillary premolar teeth was 1.40 ± 0.75 mm in the region of the Alveolar crest, 1.28 ± 0.80 mm at the medial area of the radix, and 1.84 ± 1.16 mm at the apex. Maxillary molar teeth showed a BBT of 1.42 ± 0.74 at the alveolar crest, 1.56 ± 1.05 in the middle part, and 2.78 ± 2.04 mm at the apex. In the mandible, the value distributions were: 0.95 ± 0.58 mm between the crestal part of the root and the surface of the buccal bone, 0.92 ± 0.66 mm at 4 to 9 mm apically to the alveolar crest, and 2.90 ± 1.58 mm at the apex of the radix. More posteriorly located teeth took on slightly higher values with a crestal thickness of 0.86 ± 0.0.51 mm and value distributions of 1.18 ± 0.70 mm at the medial area of the radix and 2.97 ± 1.56 mm at the apex. Mandibular molar teeth showed a crestal BBT of 1.20 ± 0.96 mm, and at 4 to 9 mm apically to the alveolar crest, BBT was 2.62 ± 2.02 mm and 5.17 ± 3.23 at the apex of the tooth (Figure 2, Figure 3 and Figure 4 and Table 2).

Note that the means and the standard deviations from the simulated aggregated region samples naturally slightly deviate from those obtained by (1) and (2). Based on the simulation results, the portion of patients showing a BBT of less than 1 mm were calculated. The results are displayed in Table 2.

## 4. Discussion

While the original treatment protocol required fully healed alveolar ridges prior to implant placement, in the 1990s these protocols were modified towards implant insertion in fresh extraction sockets [2]. As immediate dental implant placement has been a subject of great interest over the last few years, the clinical premises for this approach have been controversially discussed. One of those heavily discussed premises is a facial bone wall of at least 1 mm in thickness [4]. Hence, this study aimed to systematically review studies analyzing BBT in healthy patients in order to check the feasibility of this procedure in daily practice.

This review was able to show that in the analyzed population the BBT of maxillary front-teeth at 1 to 9 mm to the alveolar crest was on average smaller than 1 mm. The same applies to the mandibular anterior teeth. Regarding the high standard deviation, it can be assumed that in more apically located regions of the anterior mandible, too, bone thickness was smaller than 1 mm in a relevant portion of cases. At the apex of frontal teeth, the average bone thickness was >1 mm. More posteriorly-located teeth were shown to have an average bone thickness of more than 1 mm, both in the mandible and maxilla. Premolars and molars in the maxilla nevertheless showed only an average BBT of <2 mm; only bone thickness in the molar apex region had values of 2.8 mm. In contrast, the mean apical BBT was greater than 2 mm already in the anterior region of the mandible; here, the BBT increased significantly from mesial to distal. This corresponds to a study by Schwartz-Arad et al. who reported a 5-year cumulative survival rate of 89% in immediate implantation with a better prognosis in the mandible compared to maxillary placed implants, especially in the posterior part of the jaw (molar region) [118]. Those findings suggest that caution is needed in performing immediate implantation in the front teeth region. Not only can the bone thickness be reduced, but the buccal bone wall can also be damaged during surgery. In a study by Chen et al. of 34 extracted maxillary central incisors, 52% demonstrated defects of the buccal bone wall [27]. Cooper et al. reported of 21% of cases showing significant bone loss after extraction [119]. The absence of the buccal bone wall can result in aesthetic problems, an increase of stress in the coronal portion of the implant subjected to loading, peri-implant pockets, bacterial colonization, or the development of peri-implant disease [120]. Values of less than 1 mm BBT were correlated to a vertical resorption of 0.21–1.85 mm depending on the type of prosthetic connection [3,121,122]. A minimum thickness of 2 mm was advocated in order to avoid resorption and maintain soft tissue [120,123]. Bone resorption might be caused by a lack of blood supply or the surgical trauma of the implant placement, but can also be influenced by other factors, such as peri-implant soft tissue height, implant design, placement level, and the position and timing of the abutment connection [120]. As a result of bone remodeling, bone resorption occurs six months after tooth extraction, and greater loss is observed in the buccal bone plate [124]. Koh et al. reported that 50% of the original BBT undergoes resorption after immediate implant placement [5], whereas some studies suggested that immediate placement can reduce bone resorption [124]. In consideration of anticipated bone resorption, the implant shoulder should be placed just apical to the mid-facial bone crest to compensate for 0.5 to 1.0 mm of crestal bone loss [3].

It is subjected that bone stability is related to the thickness of the bone at the time of implant placement [120]. Sites with facial bone walls >2 mm were shown to have a better bone fill after immediate implant placement than sites with a thin buccal bone wall (<1 mm) [45]. In addition to hard tissue resorption, greater mucosal recession occurred in implant sites with less than 1 mm of BBT [120] with pre-existing defects of the buccal bone, thin soft tissue biotype, and facial malposition of the implant [2].

Immediate implant placement offers some advantages over delayed surgical protocols: Mello et al. reported a reduction of time required for osseointegration, a minimization of bone resorption by maintaining the periodontal architecture, as well as superior aesthetic results, especially in the front teeth region [6,7,124]. In contrast, other studies did not show any beneficial effect of immediate implant placement on dimensional reduction of alveolar bone and buccal bone loss [125,126]. Immediate implant placement is not only supposed to preserve the alveolar ridge, decrease morbidity and rehabilitation time, as well as to increase patient satisfaction [5,6,7,71,124,127,128,129,130,131], but is also supposed to be more cost-efficient [5,6,124] and believed to offer psychological benefits [5,6]. Yet, there are also some disadvantages associated with immediate implantation, such as lower implant survival rates, marginal bone loss, and the affection of peri-implant soft tissue [5,7,132,133]. The unpredictability of hard and soft tissue changes following immediate implant placement is a key factor that needs to be considered when immediate implant placement is taken into account. Generally, implant primary stability is difficult to achieve, as implants usually do not have direct contact with the alveolar bone. Furthermore, bone graft/membrane is often needed [3,5]. In a Cochrane systematic review, Esposito et al. concluded that while immediate placed implants may be at a higher risk of implant failure and complications, the aesthetic outcome might be superior [134]. In accordance with this, Chrcanovic et al. concluded that immediate implant placement affects implant failure rates but does not affect marginal bone loss or the occurrence of postoperative infection [6]. They suggested that the observed difference regarding the failure rates can be attributed to critical primary stability, as implants usually do not have direct contact with the alveolar bone [6].

Since many healthy patients are assumed to have a BBT smaller than 1 mm, cautious clinical and radiographic assessment is mandatory. This especially applies to patients suffering from diseases that cause a reduced amount of bone.

The effect of age-related hormonal changes within the jaw is still largely unexplored. Micro CT scans show changes of bone structure in postmenopausal women. These changes are associated with bone turnover markers related to bone loss. In a retrospective analysis of 239 individuals, Zhang et al. showed significant differences both between pre- and postmenopausal women as well as between postmenopausal women and older men. Whereas in women this effect might be explained by a reduction of estrogen dependent bone remodeling after the menopause, in men it might rather be attributed to a continuous bone loss caused by a lower calcium absorption, as well as a reduced physical and decreased gonadal activity [74]. Other studies analyzing patients of different ages did not show any difference in BBT [37,47]. Neither could BBT be shown to vary by the location of measurement [37,49] or by the patients’ ethnicity or sex [47]. The latter is generally not believed to be associated with the BBT [38,47,49,63,115,135]. On the contrary, some studies show statistically significant associations between the BBT and age or sex, even if in just some sites [56,136]. Zekry et al. report contradictory findings indicating that an increase in age might be correlated with an increase in BBT. [63]. The divergent results regarding age- and sex-related differences in BBT can possibly be attributed to the subpopulation analyses conducted by Zhang et al. who divided older patients with respect to their sex. Other studies analyzed variations between patients of different sex or age but did not stratify the groups. This might have masked the impact of age-related hormonal changes in postmenopausal women.

Another factor found to affect the BBT is the facial type [137,138]. In a retrospective study consisting of 155 individuals, Ozdemir et al. reported significantly lower values in high-angle patients than in normal and low-angle individuals [138]. Furthermore, Yu et al. found a reduced thickness of the buccal bone in skeletal class III patients with facial asymmetry on the deviated side [139]. Gingival thickness was demonstrated to be significantly correlated with the thickness of the underlying bone [140,141]. In a study consisting of CBCT scans of 144 individuals, Amid et al. reported a greater BBT at 2 to 6 mm apical to the CEJ in patients with thick gingival biotype [24]. Digregorio et al. investigated the effect of rapid maxillary expansion on BBT in mixed and permanent dentitions. They found a reduction of 0.73 to 1.25 mm in thickness when the maxillary permanent first molars were used as anchorage. With regard to an incidence of dehiscence at the maxillary permanent first molars of 2.5% to 55% after rapid maxillary expansion and similar results in mixed dentition to those observed when permanent teeth were used as anchorage, the use of deciduous teeth might serve as an alternative to avoid BBT reduction [142].

Radiographs should be taken prior to implant placement as implant material may lead to a misdiagnosis of peri-implant bone thickness. Vanderstuyft et al. found an artificial increase of implant diameter in CBCT scans of 12 to 15% due to blooming artefacts and an underestimation of peri-implant bone thickness of 0.3 mm. Within the transition zone of additional 0.45 mm around the implant, the buccal bone cannot always be seen [143].

Rédua et al. analyzed the correlation between spiral CT and CBCT with similar voxel sizes and found a significant correlation for direct measurements of the alveolar bone height [83]. More relevant than the image technique seems to be the real bone thickness. Rédua et al. found an absolute error that was smaller than 1 mm in measurement sites thicker than 0.6 mm. When the bone thickness was smaller than 0.6 mm, the measurements showed great variation for both CBCT and CT scan. The mean difference between real thickness and measurements taken by (CB-)CT was 0.03–0.28 mm in bones thicker than 6 mm and around 1.84–1.89 mm in bones thinner than 0.6 mm. They concluded that the BBT tends to be overestimated by (CB-)CT scans irrespective of the modality [83]. These results strengthen the assumption that a relevant proportion of healthy individuals have a BBT of <1 mm. Furthermore, they support the need of a critical selection of indication for immediate dental implant placement.

The present systematic review is limited to publications in English, German, French, and Chinese. Its validity might thus be undermined by possibly missing relevant articles. Most included studies were retrospective clinical trials; the risk of bias analysis showed that most included studies have a high risk of bias due to the lack of randomization and blinding.

The number of included patients varied considerably among papers. Unfortunately, just one paper (Temple, Schoolfield et al. 2017) provided us with all their measured data. Rather, most of them included information on the BBTs at the respective measurement point only in the form of summary statistics, most importantly the arithmetic mean and standard deviation. Yet, these two values vary substantially among papers, even if they relate to the same measurement point.

Furthermore, even upon request, only two papers (Temple, Schoolfield et al. 2017 and Amid, Mirakhori et al. 2017) disclosed how many BBTs were measured at every single measurement point that was covered in the respective paper. Therefore, the exact number of measured BBTs is unknown. When trying to estimate this number, one needs to take into account two opposing effects. On the one hand, based on Temple, Schoolfield et al. (2017) and Amid, Mirakhori et al. (2017), it seems to be that in the majority of papers the BBTs of all included patients were not measured at every single measurement point that was considered in that paper. Therefore, the number of patients per measurement point is usually significantly smaller than the total number of patients included. For instance, the data Temple, Schoolfield et al. provided us included altogether 171 patients and BBTs at 24 distinct measurement points. However, not a single patient’s BBT was measured at more than eight measurement points. Rather, the 171 patients were distributed over the 24 measurement points, yet not in a uniform way. While the BBTs of the first premolars of the mandible were measured in 66 patients, the BBTs of the first molar of the mandible were measured in merely 22 patients. Amid et al. had even bigger differences among the patient numbers at their included measurement points. While the canines of the maxilla contained BBTs of only 16 patients, the lateral incisors of the maxilla included those of 171 patients. On the other hand, there are also papers that report a higher number of measured teeth than the total number of BBTs one obtains, assuming that all patients had their BBTs measured at every included measurement point. This can only be explained in a way that the measurement points of the right and left side of the jaw were pooled. However, we lack the necessary information to infer the exact number of patients per measurement point. We therefore did all calculations assuming that the number of patients per measurement point was equal to the number of patients in that study, resulting in a number of 38,840 measured BBTs. Likely, this results in a small overestimation of the actual number of measurement point samples/BBTs. Fortunately, if the proportion of the actually included patients per measurement point to the total number of patients included in a study follows a random mechanism, our obtained results will still be unbiased. Yet, to account for the resulting uncertainty, the standard errors of all subsequent estimates are substantially scaled up.

Moreover, a limitation of this study is, of course, that buccal bone thickness was measured only on tooth-bearing segments of the jaw and not on extraction sockets. Measuring the buccal bone thickness after tooth extraction gives an even better impression of the feasibility of immediate implant placement according to the established limits. Obviously, a further reduction of the bone thickness due to the extraction of the tooth must be taken into account if a prognosis on the feasibility of the procedure is to be made on the basis of (CB)CT data. In this respect, the collection of data on buccal bone thickness after tooth extraction (ideally measured directly and not by radiographic imaging) would be desirable but is not practical due to the considerably smaller number of studies on this topic and their heterogeneity.

## 5. Conclusions

This study aimed to systematically review studies analyzing BBT in healthy patients in order to check the feasibility of immediate implant placement in daily practice. The review showed an average BBT of <1 mm in over 60% of the cases, in both the maxilla and mandible front tooth region. In contrast, posteriorly located areas showed thicker buccal bone. Since one can assume that in a relevant portion of healthy and even more in diseased patients the BBT is less than 1 mm, careful pre-surgical assessment and critical selection of indication is required to achieve the best functional and aesthetic outcome when using immediate implant placement.

## Figures and Tables

**Figure 1 healthcare-09-01663-f001:**
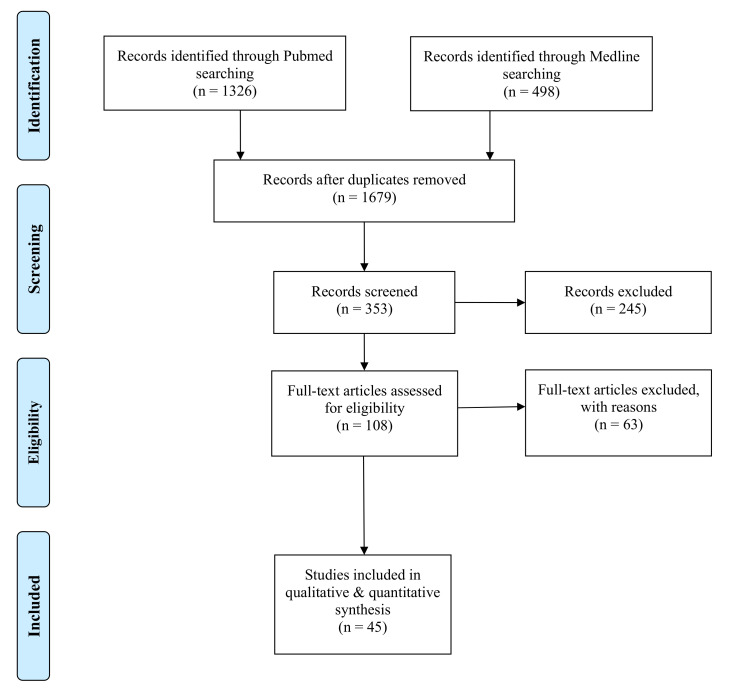
PRISMA flow diagram.

**Figure 2 healthcare-09-01663-f002:**
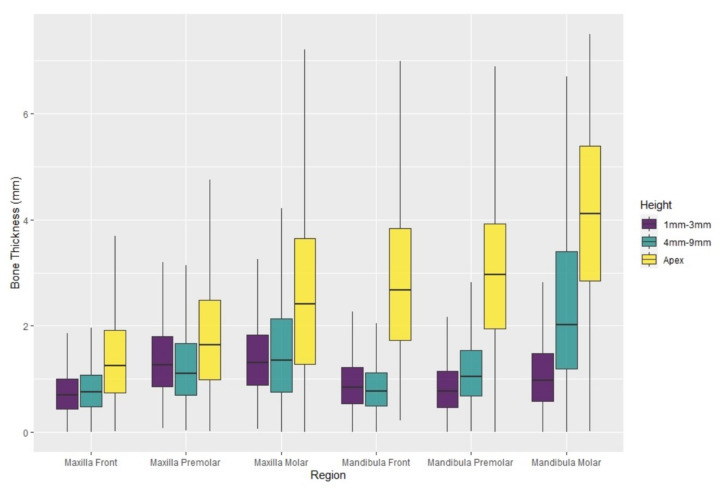
Boxplot diagram. Shown are the data plots for the six regions analyzed. The different measuring points were located at 1–3 mm and 4–9 mm apical to the alveolar crest and in the region of the tooth apex. On the one hand, the increase of the buccal bone thickness (BBT) from crestal to apical, as well as from frontal to posterior, is displayed. The bone lamella in the mandible is also significantly thicker than in the maxilla, especially in the apical regions.

**Figure 3 healthcare-09-01663-f003:**
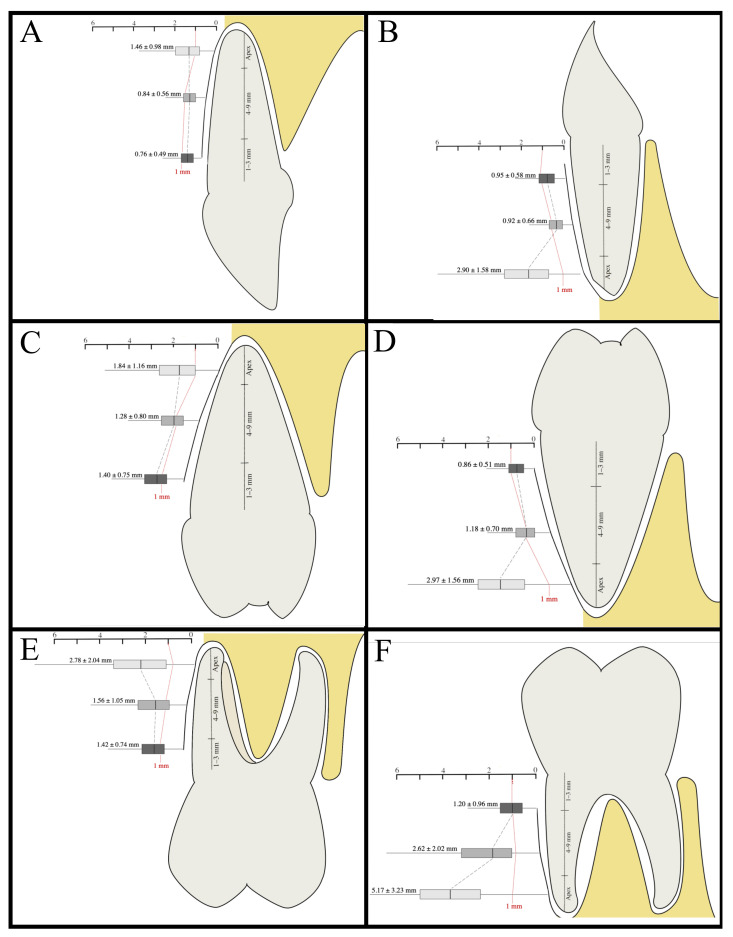
Buccal bone thickness at different regions. This figure shows (1) the measurement region as height between the tooth apex and the alveolar crest and (2) the average bone thickness as measurements taken between the respective tooth and the outer surface of the buccal bone at different regions of the radix. (**A**). Buccal bone thickness in maxillary frontal teeth. (**B**). Buccal bone thickness in mandibular front teeth. (**C**). Buccal bone thickness in maxillary premolar teeth. (**D**). Buccal bone thickness in mandibular premolar teeth. (**E**): Buccal bone thickness in maxillary molar teeth. (**F**). Buccal bone thickness in mandibular molar teeth.

**Figure 4 healthcare-09-01663-f004:**
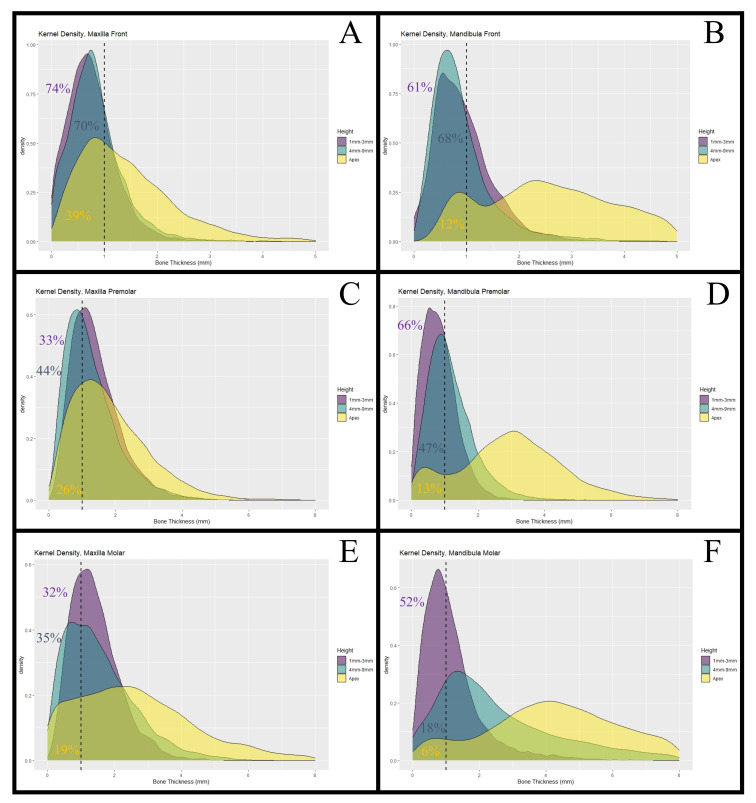
Kernel density. (**A**) Maxilla front. The figure shows the kernel density estimates (Gaussian kernel) for the simulated values of the maxillary frontal teeth (central incisor to canine). The proportions of simulated values that were smaller than 1 mm (dashed line) were: 74% at the alveolar crest (height: 1–3 mm), 70% at 4 to 9 mm from the alveolar crest, and 39% at the apex of the radix. (**C**) Maxilla premolar. The proportions of simulated values that were smaller than 1 mm (dashed line) were: 33% at the alveolar crest (height: 1–3 mm), 44% at 4 to 9 mm from the alveolar crest, and 26% at the apex of the radix. (**E**) Maxilla molar. The proportions of simulated values that were smaller than 1 mm (dashed line) were: 32% at the alveolar crest (height: 1–3 mm), 35% at 4 to 9 mm from the alveolar crest, and 19% at the apex of the radix. (**B**) Mandible anterior. The proportions of simulated values that were smaller than 1 mm (dashed line) were: 61% at the alveolar crest (height: 1–3 mm), 68% at 4 to 9 mm from the alveolar crest, and 12% at the apex of the radix. (**D**) Mandible premolar. The proportions of simulated values that were smaller than 1 mm (dashed line) were: 66% at the alveolar crest (height: 1–3 mm), 67% at 4 to 9 mm from the alveolar crest, and 13% at the apex of the radix. (**F**) Mandible molar. The proportions of simulated values that were smaller than 1 mm (dashed line) were: 52% at the alveolar crest (height: 1–3 mm), 18% at 4 to 9 mm from the alveolar crest, and 6% at the apex of the radix.

**Table 1 healthcare-09-01663-t001:** Study characteristics. List of studies included in the review for qualitative synthesis.

Study	Patients	Geographic Location	Men	Women	Teeth	Height	Maxilla (Mean ± SD)	Mandible (Mean ± SD)
Adiguzel 2017 [20]	113	Turkey	55	58	451	Apex	1st premolar: 2.12 ± 0.78 2nd premolar: 3.16 ± 0.84 1st molar: 2.47 ± 0.92 2nd molar: 3.47 ± 1.06	
Al-Jandan 2013 [21]	50	USA (PA)	22	28	250	Apex		Canine: 3.3 ± 0.84 1st premolar: 3.7 ± 0.89 2nd premolar: 3.14 ± 0.81 1st molar: 3.8 ± 0.81 2nd molar: 6.58 ± 1.27
AlMasri 2015 [22]	16	Syria	5	11	32	1 mm from alveolar crest, Apex		Central incisor: 1 (mm) 0.17 ± 0.29 (Apex) 3.64 ± 1.18
AlTarawneh 2018 [23]	120	Jordan	39	81	720	3 mm from alveolar crest, Apex	Central incisor 3 (mm) 0.73 ± 0.25 (Apex) 0.59 ± 0.29 Lateral incisor: 3 (mm) 0.69 ± 0.28 (Apex) 0.49 ± 0.29 Canine: 3 (mm) 0.74 ± 0.29 (Apex) 0.39 ± 0.30	
Amid 2017 [24]	144	Iran	60	84	621	2/4/6 mm from CEJ	Central incisor: 2 (mm) 0.35 ± 0.4 4 (mm) 0.77 ± 0.25 6 (mm) 0.75 ± 0.23 Lateral incisor: 2 (mm) 0.29 ± 0.39 4 (mm) 0.78 ± 0.36 6 (mm) 0.75 ± 0.31 Canine: 2 (mm) 0.36 ± 0.44 4 (mm) 0.76 ± 0.36 6 (mm) 0.79 ± 0.33	
Behnia 2015 [25]	18	Iran	11	7	108	1/4/8 mm from alveolar crest	Central incisor: 1 (mm) 0.78 ± 0.23 4 (mm) 0.66 ± 0.2 8 (mm) 0.69 ± 0.43 Lateral incisor: 1 (mm) 0.86 ± 0.39 4 (mm) 0.78 ± 0.54 8 (mm) 0.71 ± 0.59 Canine: 1 (mm) 0.69 ± 0.35 4 (mm) 0.77 ± 0.42 8 (mm) 0.7 ± 0.31	
Botelho 2020 [26]	87	Brazil	24	63	522	3/5/7 mm from CEJ	Central incisor: 3 (mm) 0.6 ± 0.5 5 (mm) 0.8 ± 0.4 7 (mm) 0.7 ± 0.5 Lateral incisor: 3 (mm) 0.6 ± 0.6 5 (mm) 0.9 ± 0.6 7 (mm) 0.7 ± 0.6 Canine: 3 (mm) 0.5 ± 0.5 5 (mm) 0.9 ± 0.5 7 (mm) 0.8 ± 0.6	
Chen 2017 [27]	16	Taiwan	6	5	96	Apex	Central incisor: 1.12 ± 0.45 Lateral incisor: 0.85 ± 0.49 Canine: 1.50 ± 1.49	
D’Silva 2020 [28]	66	USA (NY)	37	29	363	4 mm from CEJ	Central incisor: 0.96 ± 0.39 Lateral incisor: 1.1 ± 0.56 Canine: 1.04 ± 0.63	
Demircan 2015 [29]	60	Turkey	30	30	230	1/2/5 mm from alveolar crest	Central incisor: 1 (mm) 0.7 ± 0.17 3 (mm) 0.84 ± 0.179 5 (mm) 0.76 ± 0.21 Lateral incisor: 1 (mm) 0.78 ± 0.22 3 (mm) 0.85 ± 0.26 5 (mm) 0.69 ± 0.2	
El Nahass 2015 [30]	93	Egypt	31	62	350	1/2/4 mm from alveolar crest	Central incisor: 1 (mm) 0.72 ± 0.19 2 (mm) 0.78 ± 0.18 4 (mm) 0.81 ± 0.1 Lateral incisor: 1 (mm) 0.73 ± 0.19 2 (mm) 0.84 ± 0.25 4 (mm) 0.84 ± 0.25	
Eraydin 2018 [31]	24	Turkey	10	14	48	3 mm from CEJ, Apex		Central incisor: 3 (mm) 0.19 ± 0.38 (Apex) 4.21 ± 1.36 Lateral incisor: 3 (mm) 0.39 ± 0.57 (Apex) 4.27 ± 1.64
Farahamnd 2017 [32]	132	Tehran	65	67	792	2/5/8 mm from alveolar crest	Central incisor: 2 (mm) 0.69 ± 0.63 5 (mm) 0.66 ± 0.62 8 (mm) 0.54 ± 0.61 Lateral incisor: 2 (mm) 0.81 ± 0.64 5 (mm) 0.76 ± 0.66 8 (mm) 0.51 ± 0.53 Canine: 2 (mm) 0.9 ± 0.72 5 (mm) 0.79 ± 0.64 8 (mm) 0.58 ± 0.61	
Foosiri 2018 [33]	51	NA	21	30	306	3/6 mm from CEJ, Apex		Central incisor: 3 (mm) 0.38 ± 0.22 6 (mm) 0.8 ± 0.55 (Apex) 4.44 ± 1.27 Lateral incisor: 3 (mm) 0.49 ± 0.3 6 (mm) 1.28 ± 0.83 (Apex) 4.39 ± 1.18 Canine: 3 (mm) 1.33 ± 0.9 6 (mm) 2.25 ± 1.06 (Apex) 5.53 ± 1.44
Gakonyo 2018 [34]	184	Kenya	85	90	1104	4 mm from CEJ	Central incisor: 0.58 ± 0.4 Lateral incisor: 0.54 ± 0.42 Canine: 0.52 ± 0.47	
Ganji 2019 [35]	32	Saudi Arabia	9	7	128	3 mm from CEJ	1st premolar: 1.06 ± 0.47 2nd premolar: 1.37 ± 0.36	
Gluckman 2018 [36]	150	South Africa	67	63	591	1 mm from alveolar crest, Apex	Central incisor: 1 (mm) 0.6 ± 0.3 (Apex) 1.2 ± 0.8 Lateral incisor: 1 (mm) 0.7 ± 0.3 (Apex) 1.5 ± 1.2 Canine: 1 (mm) 0.6 ± 0.3 (Apex) 1.4 ± 1.0	
Januário 2011 [37]	250	Brazil	118	132	1500	1/3/5 mm from CEJ	Central incisor: 1 (mm) 0.6 ± 0.3 3 (mm) 0.6 ± 0.4 5 (mm) 0.5 ± 0.3 Lateral incisor: 1 (mm) 0.7 ± 0.3 3 (mm) 0.7 ± 0.4 5 (mm) 0.5 ± 0.4 Canine: 1 (mm) 0.6 ± 0.3 3 (mm) 0.6 ± 0.4 5 (mm) 0.6 ± 0.4	
Jin 2005 [38]	66	Korea	33	33	1806	Apex	Central incisor: 2.05 ± 0.49 Lateral incisor: 1.84 ± 0.47 Canine: 1.64 ± 0.45 1st premolar: 1.68 ± 0.5 2nd premolar: 1.99 ± 0.69 1st molar: 2.73 ± 1.74 2nd molar: 3.61 ± 1.11	Central incisor: 2.07 ± 0.52 Lateral incisor: 2.31 ± 0.55 Canine: 2.48 ± 0.72 1st premolar: 3.02 ± 0.91 2nd premolar: 3.68 ± 1.22 1st molar: 4.09 ± 1.25 2nd molar: 7.34 ± 1.65
Kheur 2016 [39]	150	India	NA	NA	150	3 mm from CEJ, Apex	Central incisor 3 (mm) 0.93 ± 0.39 (Apex) 1.57 ± 0.89	
Khoury 2016 [40]	47	Lebanon	16	31	282	4/6/8/10 mm from CEJ	Central incisor: 4 (mm) 1.05 ± 0.37 6 (mm) 0.97 ± 0.34 8 (mm) 0.91 ± 0.28 10 (mm) 0.91 ± 0.38 Lateral incisor: 4 (mm) 1.07 ± 0.54 6 (mm) 0.9 ± 0.63 8 (mm) 0.63 ± 0.61 10 (mm) 0.59 ± 0.7 Canine: 4 (mm) 0.95 ± 0.51 6 (mm) 0.81 ± 0.6 8 (mm) 0.66 ± 0.58 10 (mm) 0.51 ± 0.46	
Lau 2011 [41]	170	China	76	94	340	Apex	Central incisor: 2.04 ± 1.0.1	
Lee 2019 [42]	20	Korea	9	11	80	3/5 mm from CEJ, Apex	Central incisor: 3 (mm) 1.1 ± 0.3 5 (mm) 1.0 ± 0.4 (Apex) 2.3 ± 0.8 Lateral incisor: 3 (mm) 1.2 ± 0.4 5 (mm) 1.0 ± 0.4 (Apex) 2.2 ± 0.8	
Lin 2018 [43]	21	Taiwan	9	12	126	3/5 mm from CEJ	Central incisor: 3 (mm) 0.98 ± 0.48 5 (mm) 0.72 ± 0.63 Lateral incisor: 3 (mm) 0.97 ± 0.66 5 (mm) 0.42 ± 0.53 Canine: 3 (mm) 1.14 ± 0.76 5 (mm) 0.61 ± 0.71	
López-Jarana 2018 [44]	49	Spain	19	30	403	4 mm from alveolar crest, Apex	Central incisor: 4 (mm) 1.02 ± 0.49 (Apex) 1.61 ± 0.95 Canine: 4 (mm) 1.27 ± 1.95 (Apex) 1.26 ± 0.68 1st premolar: 4 (mm) 1.43 ± 0.95 (Apex) 2.19 ± 1.68 1st molar: 4 (mm) 1.55 ± 1.41 (Apex) 2.15 ± 1.68	Central incisor: 4 (mm) 0.94 ± 0.77 (Apex) 3.19 ± 1.91 Canine: 4 (mm) 1.08 ± 0.86 (Apex) 3.54 ± 1.87 1st premolar: 4 (mm) 1.49 ± 0.97 (Apex) 3.81 ± 1.83 1st molar: 4 (mm) 3.12 ± 2.03 (Apex) 6.78 ± 2.93
Matsuda 2016 [45]	95	USA (CA)	32	63	150	3 mm from alveolar crest	1st molar: 1.58 ± 0.6	
Nahás-Scocate 2014 [46]	30	Brazil	12	18	60	Apex	Central incisor: 0.99 ± 0.59	
Nowzari 2012 [47]	101	USA (LA)	53	48	202	1/2/3/4/5/6/7/8/9/10 mm from alveolar crest	Centra incisor: 1 (mm) 1.0 ± 0.3 2 (mm) 1.2 ± 0.4 3 (mm) 1.2 ± 0.4 4 (mm) 1.1 ± 0.4 5 (mm) 1.1 ± 0.4 6 (mm) 1.0 ± 0.4 7 (mm) 1.0 ± 0.5 8 (mm) 1.0 ± 0.7 9 (mm) 1.0 ± 0.7 10 (mm) 1.2 ± 1.0	
Nucera 2017 [48]	30	Italy	15	15	120	6/11 mm from CEJ		1st molar: 6 (mm) 0.25 ± 0.44 11 (mm) 2.18 ± 1.46 2nd molar: 6 (mm) 3.76 ± 2.53 11 (mm) 5.57 ± 2.42
Park 2014 [49]	20	Korea	9	11	120	3/5 mm from alevolar crest, Apex		Canine: 3 (mm) 0.7 ± 0.44 5 (mm) 0.6 ± 0.54 (Apex) 4.11 ± 1.55 1st premolar: 3 (mm) 0.73 ± 0.42 5 (mm) 0.79 ± 0.44 (Apex) 4.05 ± 1.24 2nd premolar: 3 (mm) 1.48 ± 0.72 5 (mm) 1.76 ± 0.74 (Apex) 5.15 ± 1.18
Pascual 2017 [50]	15	Spain	8	7	180	4 mm from CEJ, Apex	Central incisor: 4 (mm) 1.07 ± 0.96 (Apex) 1.69 ± 0.65 Lateral incisor: 4 (mm) 1.31 ± 0.44 Apex) 1.98 ± 0.79 Canine: 4 (mm) 1.19 ± 0.44 (Apex) 1.71 ± 0.81	Central incisor: 4 (mm) 1.21 ± 1.05 (Apex) 3.94 ± 1.79 Lateral incisor: 4 (mm) 1.23 ± 1.09 (Apex) 3.69 ± 1.58 Canine: 4 (mm) 1.47 ± 1.42 (Apex) 3.7 ± 1.64
Porto 2020 [51]	422	Brazil	28	394	1400	Apex	Central incisor: 1.59 ± 0.67 Lateral incisor: 2.3 ± 1.2 Canine: 1.49 ± 0.86 1st premolar: 1.13 ± 0.68 2nd premolar: 2.20 ± 1.21 1st molar: 1.98 ± 3.3 2nd molar: 3.51 ± 2.15	Central incisor: 2.72 ± 1.3 Lateral incisor: 3.06 ± 1.29 Canine: 3.43 ± 1.31 1st premolar: 3.27 ± 1.04 2nd premolar: 3.65 ± 1.35 1st molar: 4.45 ± 1.46 2nd molar: 6.65 ± 4.47
Ramanauskaite 2020 [52]	60	Germany	29	31	707	Alveolar crest, Apex	Incisor: (AC) 0.56 ± 0.21 (Apex) 1.4 ± 0.93 Premolar: (AC) 0.84 ± 0.42 (Apex) 1.45 ± 1.35 Molar: (AC) 0.97 ± 0.6 (Apex) 1.64 ± 1.40	Incisor: (AC) 0.55 ± 0.27 (Apex) 3.64 ± 1.63 Premolar: (AC) 0.51 ± 0.29 (Apex) 3.48 ± 1.45 Molar: (AC) 0.97 ± 1.1 (Apex) 6.7 ± 2.31
Rojo-Sanchis 2017 [53]	44	Spain	25	19	144	3/5 mm from CEJ	1st premolar: 3 (mm) 1.71 ± 0.89 5 (mm) 1.44 ± 1.00 2nd premolar: 3 (mm) 2.43 ± 0.82 5 (mm) 2.31 ± 1.06	
Sendyk 2017 [54]	35	Brazil	16	19	980	3/8 mm from CEJ	Central incisor: 3 (mm) 0.9 ± 0.2 8 (mm) 1.1 ± 0.4 Lateral incisor: 3 (mm) 0.8 ± 0.3 8 (mm) 1.0 ± 0.4 Canine: 3 (mm) 0.6 ± 0.2 8 (mm) 0.9 ± 0.3 1st premolar: 3 (mm) 0.7 ± 0.3 8 (mm) 1.0 ± 0.4 2nd premolar: 3 (mm) 1.3 ± 0.5 8 (mm) 1.5 ± 0.6 1st molar: 3 (mm) 1.0 ± 0.4 8 (mm) 1.2 ± 0.6 2nd molar: 3 (mm) 1.3 ± 0.6 8 (mm) 2.1 ± 1.0	Central incisor: 3 (mm) 0.5 ± 0.1 8 (mm) 1.5 ± 0.7 Lateral incisor: 3 (mm) 0.5 ± 0.1 8 (mm) 1.0 ± 0.5 Canine: 3 (mm) 0.5 ± 0.1 8 (mm) 0.9 ± 0.4 1st premolar: 3 (mm) 0.5 ± 0.1 8 (mm)1.2 ± 0.7 2nd premolar: 3 (mm) 0.8 ± 0.3 8 (mm) 1.8 ± 0.8 1st molar: 3 (mm) 1.0 ± 0.4 8 (mm) 2.3 ± 1.0 2nd molar: 3 (mm) 2.8 ± 1.5 8 (mm) 5.6 ± 1.6
Shrestha 2019 [55]	146	China	65	81	876	4 mm from CEJ	Central incisor: 0.89 ± 0.51 Lateral incisor: 0.85 ± 1.12 Canine: 0.84 ± 0.68	
Temple 2016 [56]	265	USA	119	146	934	1/3/5 mm apical to alveolar crest	1st premolar: 1 (mm) 0.95 ± 0.38 3 (mm) 0.81 ± 0.43 5 (mm) 0.66 ± 0.38 2nd premolar: 1 (mm) 1.46 ± 0.66 3 (mm) 1.44 ± 0.79 5 (mm) 1.18 ± 0.67 1st molar: 1 (mm) 0.98 ± 0.40 3 (mm) 0.86 ± 0.48 5 (mm) 0.76 ± 0.54 2nd molar: 1 (mm) 1.58 ± 0.67 3 (mm) 1.92 ± 0.86 5 (mm) 2.13 ± 1.08	1st premolar: 1 (mm) 0.54 ± 0.37 3 (mm) 0.61 ± 0.55 5 (mm) 0.90 ± 0.69 2nd premolar: 1 (mm) 0.66 ± 0.45 3 (mm) 0.71 ± 0.53 5 (mm) 1.08 ± 0.68 1st molar: 1 (mm) 0.61 ± 0.29 3 (mm) 1.07 ± 0.85 5 (mm) 1.65 ± 1.29 2nd molar: 1 (mm) 0.88 ± 0.68 3 (mm) 2.03 ± 1.46 5 (mm) 3.65 ± 1.67
Üner 2019 [57]	160	Turkey	80	80	320	3/6/9 mm from CEJ	Central incisor: 3 (mm) 1.19 ± 0.4 6 (mm) 1.15 ± 0.44 9 (mm) 1.06 ± 0.5	
Wang 2014 [58]	300	China	133	167	2400	4 mm from CEJ	Central incisor: 0.8 ± 0.4 Lateral incisor: 0.7 ± 0.4 Canine: 0.7 ± 0.5 1st premolar: 1.2 ± 0.6 2nd premolar: 1.7 ± 0.8	
Yoshimine 2012 [59]	30	Japan	8	22	240	Apex	1st premolar: 0.76 ± 0.57 2nd premolar: 2.13 ± 1.26 1st molar: 2.39 ± 1.47 2nd molar: 4.3 ± 2.1	
Younes 2016 [60]	21	Belgium	7	14	126	1/2/3/4/5 from alveolar crest	Central incisor: 1 (mm) 0.97 ± 0.28 2 (mm) 1.1 ± 0.38 3 (mm) 1.1 ± 0.43 4 (mm) 1.1 ± 0.41 5 (mm) 1.08 ± 0.43 Lateral incisor: 1 (mm) 0.95 ± 0.34 2 (mm) 1.21 ± 0.43 3 (mm) 1.27 ± 0.63 4 (mm) 1.22 ± 0.6 5 (mm) 1.15 ± 0.76 Canine: 1 (mm) 0.89 ± 0.24 2 (mm) 0.99 ± 0.3 3 (mm) 1.04 ± 0.4 4 (mm) 1.00 ± 0.53 5 (mm) 0.96 ± 0.57	
Yuan 2018 [61]	40	China	16	24	80	1/3/5 mm from alveolar crest	Central incisor: 1 (mm) 0.89 ± 0.30 3 (mm) 0.81 ± 0.30 5 (mm) 0.67 ± 0.28 Lateral incisor: 1(mm) 0.8 ± 0.33 3 (mm) 0.85 ± 0.46 5 (mm) 0.43 ± 0.34	
Zahedi 2018 [62]	170	Iran	69	101	1354	Apex		1st premolar: 0.85 ± 0.88 2nd premolar: 1.3 ± 1.05 1st molar: 1.43 ± 1.28 2nd molar: 5.54 ± 2.12
Zekry 2012 [63]	200	China	74	126	2400	1/3/5 mm from alveolar crest	Central incisor: 1 (mm) 0.9 ± 0.28 3 (mm) 0.89 ± 0.3 5 (mm) 0.81 ± 0.3 Lateral incisor: 1 (mm) 0.94 ± 0.34 3 (mm) 0.88 ± 0.36 5 (mm) 0.68 ± 0.29 Canine: 1 (mm) 1.09 ± 0.34 3 (mm) 1.08 ± 0.47 5 (mm) 0.84 ± 0.39 1st premolar: 1 (mm) 1.23 ± 0.40 3 (mm) 1.26 ± 0.51 5 (mm) 1.16 ± 0.49 2nd premolar: 1 (mm) 1.63 ± 0.62 3 (mm) 2.01 ± 0.83 5 (mm) 1.99 ± 0.92 1st molar: 1 (mm) 1.49 ± 0.57 3 (mm) 1.79 ± 0.78 5 (mm) 1.73 ± 0.88	Central incisor: 1 (mm) 0.96 ± 0.36 3 (mm) 0.83 ± 0.43 5 (mm) 0.8 ± 0.37 Lateral incisor: 1 (mm) 0.98 ± 0.35 3 (mm) 0.83 ± 0.4 5 (mm) 0.75 ± 0.32 Canine: 1 (mm) 0.99 ± 0.35 3 (mm) 0.77 ± 0.35 5 (mm) 0.68 ± 0.38 1st premolar: 1 (mm) 0.91 ± 0.33 3 (mm) 0.9 ± 0.34 5 (mm) 1.0 ± 0.38 2nd premolar: 1 (mm) 1.21 ± 0.41 3 (mm) 1.30 ± 0.48 5 (mm) 1.49 ± 0.65 1st molar: 1 (mm) 1.08 ± 0.41 3 (mm) 1.31 ± 0.52 5 (mm) 1.62 ± 0.66
Zhang 2015 [64]	105	China	69	46	1260	1 mm from alveolar crest, Apex	Central incisor: 1 (mm) 0.95 ± 0.39 (Apex) 0.86 ± 0.23 Lateral incisor: 1 (mm) 0.76 ± 0.38 (Apex) 0.9 ± 0.29 Canine: 1 (mm) 0.67 ± 0.42 (Apex) 0.98 ± 0.31	Central incisor: 1 (mm) 1.55 ± 0.52 (Apex) 0.87 ± 0.25 Lateral incisor: 1 (mm) 1.53 ± 0.55 (Apex) 0.91 ± 0.29 Canine: 1 (mm) 1.77 ± 0.6 (Apex) 0.81 ± 0.27

The following pieces of information were extracted from the articles: Number of patients analyzed, geographic location, patients grouped in men and women, number of teeth analyzed, measurement area as height between CEJ/alveolar crest and apex of the tooth, and measurement values classified into regions (maxilla, mandible). The values are displayed as mean ± SD subclassified into the tooth group and measurement height. Abbreviations: CEJ–cemento-enamel junction; SD–standard deviation.

**Table 2 healthcare-09-01663-t002:** Buccal bone thickness.

Jaw	Teeth	Height [mm]	Proportion < 1 mm	Standard Deviation	Mean Thickness [mm]	Standard Deviation [mm]
Maxilla	Central incisor–canine	1–3	74.4%	0.8%	0.76	0.49
		4–9	70.1%	1.2%	0.84	0.56
		Apex	38.5%	1.2%	1.46	0.98
	1st premolar–2nd premolar	1–3	33.1%	1.5%	1.40	0.75
		4–9	438%	2.4%	1.28	0.80
		Apex	25,5%	1.8%	1.84	1.16
	1st molar–2nd molar	1–3	31,6%	2.0%	1.42	0.74
		4–9	35.3%	2.4%	1.56	1.05
		Apex	19.1%	1.5%	2.78	2.04
Mandible	Central incisor–canine	1–3	61.2%	1.9%	0.95	0.58
		4–9	68.3%	2.2%	0.92	0.66
		Apex	12.3%	0.8%	2.90	1.58
	1st premolar–2nd premolar	1–3	66.3%	1.9%	0.86	0.51
		4–9	46.7%	2.4%	1.18	0.70
		Apex	13.1%	0.9%	2.97	1.56
	1st molar–2nd molar	1–3	51.9%	2.3%	1.20	0.96
		4–9	18.2%	2.0%	2.62	2.02
		Apex	6.4%	0.7%	5.17	3.23

This table shows the average bone thickness displayed as mean (M) and standard deviation (SD) at different regions of the radix (height). The estimated portion of patients with a buccal bone thickness smaller than 1 mm is displayed in the last column. Teeth were grouped into front (central incisor to canine) and premolar teeth and molar teeth within the maxilla and mandible.

## Data Availability

The data that support the findings of this study are available in the National Center for Biotechnology Information at https://www.ncbi.nlm.nih.gov/ (accessed on 20 December 2020). Data derived from the resources mentioned in Table 2 are available in the public domain.

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
