# Peer review of "Buccal Bone Thickness in Anterior and Posterior Teeth—A Systematic Review"

_healthcare, 2021, doi:10.3390/healthcare9121663_

Round 1

Reviewer 1 Report

Dear Authors

The manuscript is interesting, but in my opinion has several points to be improved, before it be eligible for publication .

Why language restriction? I am sure many english written articles were excluded;

In M&M section, some information should be on the results section; this section does not correspond to PRISMA guidelines; some criteria are confusing to understand; are you center on patient, socket or measurement point?

BBT is not so relevant to implant survival rates, but instead, on the need to graft or not graft the socket, and to any aesthetic problem due to it, but it is not mentioned nor discussed about it on the manuscript;

If the objective the review is to relate BBT to implant survival rate, i suggest you to compare the bone on the residual socket, where  usually periodontists place implants on immediate treatments;

I you got enough data, why not do a Meta analysis with that?

Best regards

Author Response

Thank you very much for the multitude of constructive comments on our work.

Reviewer Comment 1: Why language restriction? I am sure many English written articles were excluded.

Answer: We did include all articles written in English, German, French and Chinese; Accordingly, we do not assume that articles in English language were forgotten.

Reviewer Comment 2: (A) In M&M section, some information should be on the results section; this section does not correspond to PRISMA guidelines; (B) some criteria are confusing to understand; are you center on patient, socket or measurement point?

Answer: (A) We changed the order according to the new PRISMA checklist from 2020; (B) We are not sure what the reviewer is intending to say with this comment. If he is referring to the question of what the primary objective of this review was, then the question should be answered as follows: The focus of this review was to analyze buccal bone thicknesses at various points of the socket.

Reviewer Comment 3: (A) BBT is not so relevant to implant survival rates, but instead, on the need to graft or not graft the socket, and to any aesthetic problem due to it, (B) but it is not mentioned nor discussed about it on the manuscript.

Answer: (A) We agree with the reviewer, that BBT is not the most relevant factor for implant survival rates and that sockets with thinner bone walls require grafting procedures. But, since many patients wish for immediate implant placement and this procedure is becoming increasingly popular, we aimed to systematically review studies analyzing BBT in healthy patients in order to check the feasibility of this procedure in daily practice. One of the heavily discussed premises is a facial bone wall of at least 1 mm thickness (ITI Consensus Conference). (B) We refer to this problem within the discussion section as can be seen in lines 376–379 “The absence of the buccal bone wall can result in aesthetic problems, an increase of stress in the coronal portion of the implant subjected to loading, peri-implant pockets, bacterial colonization, or the development of peri-implant disease” and 415–421 “Yet, there are also some disadvantages associated with immediate implantation, such as lower implant survival rates, marginal bone loss and the affection of peri-implant soft tissue. The unpredictability of hard and soft tissue changes following immediate implant placement is a key factor that needs to be considered when immediate implant placement is taken into account.”

Reviewer Comment 4: If the objective the review is to relate BBT to implant survival rate, i suggest you to compare the bone on the residual socket, where usually periodontists place implants on immediate treatments;

Answer: As described earlier, the aim of this review was to investigate the requirements for immediate implant placement at different sites of the jaw. In principle, it would be useful to investigate the buccal bone thickness with a tooth already extracted; unfortunately, only very few data are available for this. In order to establish a large data set of buccal bone thicknesses at different sites, we decided to analyze CBCT and CT data of healthy patients. The inherent reduction in bone thickness due to the tooth extraction process must of course be taken into account, as well as the expected resorption of bone after implant placement, which significantly depend on the residual buccal bone thickness. We added a section discussing this (see “study limitations” in the “discussion section”).

Reviewer Comment 5: You got enough data, why not do a Meta analysis with that?

Answer: To our best knowledge, a meta-analysis requires a control and a test group. Since we only measured buccal bone thickness in healthy people (one group), no meta-analysis could be performed.

Reviewer 2 Report

The authors have developed a systematic review dealing with bone thickness in the oral cavity and the use of implants.

The manuscript is interesting but has some formatting deficiencies and mismatched search information. More specifically:

INTRODUCTION
- The introduction is very short. Authors are recommended to deepen the contents and provide epidemiological data that justify the development of this manuscript.
- The research objective is poorly stated. The goal cannot be “to conduct a systematic review”. The verb must be a concrete action in relation to the study problem.

MATERIAL AND METHODS
Search strategy:
- The search equation is wrong. The figures of the "results" do not match when the search is reproduced.

Inclusion and Exclusion Criteria:
- The authors say they only used clinical trials and then exclude other designs:
o Were doctoral theses with intervention studies included?
o Were editorials included?
o What languages were the articles eligible for?
- What was the chronological criterion? Were all items eligible through December 2020? Why haven't you updated your search in 2021?

Statistics:

The Statistics section is a mess since it seems that they talk about results, it also seems like a discussion as well as an introduction in which they propose objectives. This section should objectively describe the tools used.

RESULTS
- When entering the search equation literally, 2416 results appear in Pubmed. These figures do not coincide with the information provided in the manuscript. Authors should check for problems and simplify the approach to supplemental material.

Quality Assessment:
- This subsection does not go here, but in “Material and Methods”.

DISCUSSION
- The authors have not used "study limitations".

REFERENCES
- There are many misprints and incomplete references. Authors should review this section.
- References must go to APA or Vancouver regulations. The authors seem to have used APA format but they only respect it when citing the authors. From the title, in each reference, they use their own format.

Author Response

Thank you very much for the multitude of constructive comments on our work.

INTRODUCTION

Reviewer Comment 1: The introduction is very short. Authors are recommended to deepen the contents and provide epidemiological data that justify the development of this manuscript.

Answer: The introduction has been expanded according to the reviewer’s comments as can be seen in lines 32–38 and 56–75.

Reviewer Comment 2: The research objective is poorly stated. The goal cannot be “to conduct a systematic review”. The verb must be a concrete action in relation to the study problem.

Answer: We changed this according to the reviewer’s comment (see lines 76–81).

MATERIAL AND METHODS

Search strategy:

Reviewer Comment 3: The search equation is wrong. The figures of the "results" do not match when the search is reproduced.

Answer: We are very sorry about this error. We forgot to explicitly mention that the search was limited to human studies. Accordingly, the filter "Humans" in the "Species" section in pubmed was used as a filter to further narrow the search. This results in a number of 1,422 hits between 1984 and 2020 (12/31/2020). We are very grateful that the reviewer noticed this error and added this information in the M&M section (search strategy).

Inclusion and Exclusion Criteria:

Reviewer Comment 4: The authors say they only used clinical trials and then exclude other designs:
o Were doctoral theses with intervention studies included?
o Were editorials included?
o What languages were the articles eligible for?

Answer: Doctoral theses with intervention studies that were listed in PubMed or Medline due to publication in an international peer-reviewed journal were included; due to the restriction to the electronic data bases PubMed and Medline, no other sources have been used; we did not include editorials. The languages were: English, German, French and Chinese. We have added to the information in the "exclusion criteria" accordingly.

Reviewer Comment 5: What was the chronological criterion? Were all items eligible through December 2020? Why haven't you updated your search in 2021?

Answer: The search was conducted only until the end of 2021 for two reasons: 1. Data evaluation ended in December 2021, but nevertheless the development of the statistics took a lot of time. This was followed by an extensive revision process by all authors, resulting in the submission 10 months after the completion of the data synthesis. 2. However, it cannot be assumed that an extension of the data analysis to the year 2021 will lead to a significant change in the data, since the currently included collective also already contained 25,452 measurement points. In this respect, it can be assumed that the data, which is now 11 months out of date, represents a sufficient basis for answering the research question and that there should be no significant change in the data generated even if the data from 2021 is added.

Statistics:

Reviewer Comment 6: The Statistics section is a mess since it seems that they talk about results, it also seems like a discussion as well as an introduction in which they propose objectives. This section should objectively describe the tools used.

Answer: We have significantly shortened the statistics section. Unfortunately, due to the statistical method developed specifically for this study, the detailed description of the calculation basis is indispensable to allow a correct reconstruction of the study methods.

RESULTS

Reviewer Comment 7: When entering the search equation literally, 2416 results appear in PubMed. These figures do not coincide with the information provided in the manuscript. Authors should check for problems and simplify the approach to supplemental material.

Answer: We are very sorry about this error. We forgot to explicitly mention that the search was limited to human studies. Accordingly, the filter "Humans" in the "Species" section in PubMed was used as a filter to further narrow the search. This results in a number of 1,422 hits between 1984 and 2020 (12/31/2020). We are very grateful that the reviewer noticed this error and added this information in the M&M section (search strategy, lines 90–92).

Quality Assessment:

Reviewer Comment 8: This subsection does not go here, but in “Material and Methods”.

Answer: We changed this according to the reviewer comment.

DISCUSSION

Reviewer Comment 9: The authors have not used "study limitations".

Answer: We added a section discussion the limitations of our study (see lines 493–543).

REFERENCES

Reviewer Comment 10: There are many misprints and incomplete references. Authors should review this section.

Answer: After changing the citation style, the references should now be displayed correctly.

Reviewer Comment 11: References must go to APA or Vancouver regulations. The authors seem to have used APA format but they only respect it when citing the authors. From the title, in each reference, they use their own format.

Answer: We changed this according to the reviewer’s comment (APA).

Round 2

Reviewer 1 Report

Dear Authors

The manuscript has improved and now i will suggest it to be published.

Thank you for your considerations

Best regards

Author Response

Dear Reviewer,

we are very grateful for the many helpful suggestions. We believe they have been crucial in improving the manuscript. Thank you very much for your work.

Best regards